# Highly Conductive PDMS Composite Mechanically Enhanced with 3D-Graphene Network for High-Performance EMI Shielding Application

**DOI:** 10.3390/nano10040768

**Published:** 2020-04-16

**Authors:** Dongyi Ao, Yongliang Tang, Xiaofeng Xu, Xia Xiang, Jingxia Yu, Sean Li, Xiaotao Zu

**Affiliations:** 1School of Physics, University of Electronic Science and Technology of China, Chengdu 610054, China; aodongyi@std.uestc.edu.cn (D.A.); x.f.xu@outlook.com (X.X.); xiaxiang@uestc.edu.cn (X.X.); jxyu@uestc.edu.cn (J.Y.); 2School of Physical Science and Technology, Southwest Jiaotong University, Chengdu 610031, China; 3School of Materials Science and Engineering, University of New South Wales, Sydney 2052, Australia; sean.li@unsw.edu.au

**Keywords:** 3D graphene network, high electrical conductivity, PDMS composite, EMI SE

## Abstract

A highly conductive three-dimensional (3D) graphene network (GN) was fabricated by chemical vapor deposition on a 3D nickel fiber network and subsequent etching process. Then a lightweight and flexible polydimethylsiloxane (PDMS)/GN composite was prepared by a vacuum infiltration method by using the graphene network as a template. The composite showed the superior electrical conductivity of 6100 S/m even at a very low loading level of graphene (1.2 wt %). As a result, an outstanding electromagnetic interference (EMI) shielding effectiveness (SE) of around 40 and 90 dB can be achieved in the X-band at thicknesses of 0.25 and 0.75 mm, respectively, which are much higher than most of the conductive polymers filled with carbon. The 3D graphene network can also act as a mechanical enhancer for PDMS. With a loading level of 1.2 wt %, the composite shows a significant increase by 256% in tensile strength.

## 1. Introduction

With the booming of electronic devices and equipment over past decades, severe electromagnetic interference (EMI) and radiation pollution are brought out [1,2,3], which can easily interfere with electronic devices and also be harmful to human health [4,5]. Generally, metals with high electrical conductivity, such as copper, nickel, silver and aluminum, show good shielding performance because of the reflection of energy caused by impedance mismatch of the material and the free space [6,7,8,9]. However, metal materials suffer from the disadvantages of heavy weight, high cost, easy corrosion, and are hard to process. Recently, conductive polymer composites (CPCs) have been intensely developed for EMI shielding applications, such as automobiles, flexible electronics and wearable devices, where lightweight and flexible materials are required [10,11,12,13,14,15,16,17,18]. Carbon materials such as carbon nanotubes, graphene, and carbon nanofibers are intensely developed as the fillers in CPCs because of their excellent electrical conductivity and stability [19,20,21].

The loading level of carbon materials in CPCs is usually high to achieve the required electrical conductivity for EMI applications, because of the poor dispersion and connectivity of separated carbon. For examples, Zeng et al. reported that 76.2 wt % multiwall carbon nanotubes (MWCNT)-loaded porous PDMS (polydimethylsiloxane) composites has EMI shielding effectiveness (SE) of 21 dB in the X-band frequency range [22]. The high loading level of carbon not only wastes the expensive carbon materials but also degrades the mechanical properties of CPCs. On the other hand, a low loading level of carbon may enhance the mechanical properties of CPCs [23,24]. For example, Mahapatra et al. reported that the breaking stress and modulus of polyurethanes were enhanced by 29% and 42% with 1 wt % incorporated MWCNT [24]. The thickness is another factor that needs to be considered when preparing shielding materials. It is well established that greater thickness will lead to higher EMI SE [25]. Usually, several millimeters CPCs is required to meet the basic requirement for EMI SE applications. For example, Yan et al. reported that a 2.5 mm porous PS (polystyrene) composite with 30 wt % loading of a few layers of graphene nanosheets has an EMI SE of approximate 29.3 dB in the X-band [26]. However, these CPCs are too thick to use as thin, protective layers for EMI shielding of sensitive instruments.

In order to balance the EMI SE and mechanical properties, the reduced loading level of CPCs is necessary. An effective way is to use three-dimensional (3D) connective carbons (foam, sponge, aerogel) as the filler. The stable and high-quality 3D connective carbons can act as the electron channels to improve the electrical conductivity of CPCs. Two types of 3D connective carbon have recently been intensely developed [27,28,29,30]. One is r-GO (reduced graphene oxide) sponge with high defect density, which has poor electrical conductivity and limits its potential application in EMI shielding [31]. The other is graphene foam deposited on commercial nickel foam by the chemical vapor deposition (CVD) method. Owing to its low defect density, the conductivity of CVD graphene is much higher than that of r-GO. However, the CVD graphene foam was fragile and collapsed after nickel was etched, because the ultrahigh porosity derived from nickel foam degraded the connectivity. Consequently, the conductivity of the CPCs based on CVD graphene foam is still not high enough (lower than 500 S/m) [32]. In addition, the EMI SE of these CPCs is usually below 30 dB with a relatively large thickness of 1 mm. For example, Chen et al. reported PDMS composites based on CVD graphene foam with EMI SE of 28.91 dB at 12 GHz [33]. This limits their application in areas such as military, high-frequency PCBs (printed circuit boards), and microwave shielding rooms where high EMI SE is required. 

Herein, a highly conductive 3D graphene network (GN) was prepared on nickel networks with a porosity of ~50% using a CVD method. With this low porosity, GN grown on the template is stable and highly conductive after nickel was etched. The PDMS was infiltrated into the GN in a vacuum to prepare the composites with excellent conductivity (6100 S/m) and good flexibility. An outstanding EMI SE of about 90 dB in the X-band was achieved for the composite with a thickness of 0.75 mm at a low graphene content of 1.2 wt %. For the composite with a thickness of 0.25 mm, the EMI SE is still higher than 40 dB. In addition to the superior EMI shielding performance, the composite also shows a significant increase in mechanical performance compared with pristine PDMS.

## 2. Materials and Methods

### 2.1. Materials

The PDMS was obtained from Dow Corning Corporation (Midland, MI, USA). The argon, hydrogen, and methane gases with 99.999% purity were obtained from the National Institute of Measurement and Testing Technology (Chengdu, China). The nickel fibers with a diameter of 8 μm were purchased from Xi’an Filter Metal Materials Corporation (Xi’an, China). 

### 2.2. Preparation of Polydimethylsiloxane/Graphene Network (PDMS/GN) Composite

The preparation process of the composite is illustrated in Figure 1a. The nickel fibers were soaked in 10% HCl at 80 °C for 2 h, and then cleaned ultrasonically in alternate baths of acetone, ethanol and deionized water to remove the surface contaminants and impurities. Then nickel fibers were cut short (0.5–1 mm in length) and stirred for 1 hour in the deionized water. The obtained nickel fibers were pressed into a 3D connective network after drying in an oven at 80 °C under a pressure of 15 MPa. A cuboid 3D connective nickel network was finally prepared with a porosity of ~50% and a thickness of 0.25 mm.

The nickel network was placed in a ceramic crucible inside a quartz tube for the growth of graphene by a CVD method under ambient pressure. In brief, the nickel network was heated up to 1100 °C and kept for 20 min in mixed gas of Ar (300 sccm) and H_2_ (10 sccm). Then, CH_4_ with a flow rate of 5 sccm was introduced into the quartz tube for 20 minutes to grow graphene on the surface of nickel fibers. Finally, the nickel network was cooled to room temperature in the ambience of Ar (300 sccm) and H_2_ (10 sccm). Thereafter, the nickel template was etched away in 20% HCl at 80 °C for 2 days. Finally, the connective freestanding 3D graphene network (GN) with duplicated structure of nickel network was produced, as shown in Figure 1b. The obtained GN was washed several times in alternate baths of ethanol and deionized water and dried at 80 °C. Finally, the liquid PDMS was infiltrated into the GN in vacuum and then solidified to obtain the PDMS/GN composite.

### 2.3. Characterization Methods

A field-emission scanning electron microscopy (SEM, FEI Inspect F, Hillsboro, OR, USA) was employed to analyze the morphology of the prepared products. A Raman spectrometer (Witec, Ulm, Germany) equipped with a 488 nm laser was used to analyze the graphene structure. A transmission electron microscope (TEM) was used for microstructural analysis with a Double Cs-corrector FEI Titan Themis G2 60–300 microscope (FEI Inc., Hillsboro, OR, USA). A four-probe meter (Tektronix, Keithley 2400, Cleveland, OH, USA) was used to measure the conductivity by. An electric universal testing machine (Shimadzu, AG-10TA, Kyoto, Japan) was used to analyze the composite’s tensile strength. The S parameters of the PDMS/GN nanocomposites in the X-band frequency range were measured using a vector network analyzer (VNA, Agilent Technologies, E8363B, Santa Clara, CA, USA). A pair of holders 22.9 × 10.2 mm^2^ in size was used to hold the nanocomposite. The holders were connected to two coaxial to waveguide adaptors, which were connected to VNA with two coaxial cables. The incident power was set to be 1 mW.

## 3. Results and Discussion

Figure 2a exhibits the SEM image of the prepared nickel network, showing numerous nickel fibers with ~8 μm diameter and coarse surface crossed and stacked to form a 3D network structure. After the CVD process, the surface of the nickel fibers becomes smooth (Figure 2b). In addition, some nodes formed (red arrows in Figure 2b,c) after the nickel fibers were sintered at high temperature, which results in good connectivity of the nickel network. Figure 2c shows the cross-sectional image of the nickel network after the CVD process, and peel-off of graphene can be found at some fractures (blue arrows in Figure 2c), indicating the successfully growth of the graphene on the nickel fibers. Figure 2d,e shows the SEM images of the graphene network after removal of the nickel template. As shown in Figure 2d, the graphene network consisting of graphene belts duplicates the 3D structure of nickel network. The cross-section view (Figure 2e) further confirms the hollow structure of graphene belts. The hollow structure leads to a very low density (~10 mg/cm^3^) of the GN. The GN obtained will inherit the good connectivity of the nickel template, which results in the excellent conductivity of the GN.

The Raman spectrum of the GN is shown in Figure 3. The spectrum shows two peaks centered at ~1580 cm^−1^ and 2700 cm^−1^, which can be indexed as the *G* band and 2*D* band of graphene, respectively. The 2*D* peak is asymmetric and the intensity of the 2*D* peak is much lower than that of the *G* peak, indicating that multilayer graphene was obtained [34]. *D* band, which is related to structural defects and graphene edges, is almost negligible, indicating the high quality of graphene. The TEM results (Figure 4) further confirm the multilayer feature of the prepared graphene, and the thickness of the graphene is around 10 layers.

Figure 5a shows the photo of PDMS/GN composite. The composite is highly flexible and has a density of 0.8 g/cm^3^. Since the GN has a density of 10 mg/cm^3^, the GN loading level in the composite is 1.2 wt %. The SEM images of PDMS/GN composite are presented in Figure 5b. The GN is tightly bonded with the PDMS matrix and maintain their original 3D structure (red arrows in Figure 5c). Moreover, a standard four-point contact method was used to measure the conductivity of samples at room temperature. The composite has a conductivity of 6100 S/m, which is similar to that of the original GN (6300 S/m), indicating the 3D graphene network was intact during the infiltration and subsequent solidification treatment of PDMS. This superior conductivity is much higher than those of previous reported CPCs with higher carbon loading levels (Table 1) [35]. This result indicates the enormous potentiality of this composite in the application of EMI shielding. Pores in the composite are resulted from the residual air in the composite during the infiltration and subsequent solidification processes.

In addition to super electrical conductivity, good mechanical properties are also necessary for the application of EMI shielding. To investigate the influence of the GNs on the mechanical properties on the composite, an electric universal testing machine was used to measure the samples’ tensile strength according to GB/T 528-92. The speed control mode was used, and the stretching and releasing rate was adjusted to 1.5 mm/min, and composites filled with GNs and pristine PDMS were compared. Figure 6a shows the tensile strain-stress curves of PDMS and PDMS/GN. As is shown, the inclusion of 1.2 wt % GN significantly enhanced the tensile strength by 256% compared to that of pristine PDMS. However, the elongation at break was decreased to 77%. The enhancement of the tensile strength and the degradation of the elongation at break could be attributed to the following aspects: (1) the 3D structure of GN restricts the motion of PDMS molecule chains [36]; (2) graphene tubes have excellent tensile modulus and lower elongation at break. Lots of energy can be absorbed by GN when the strain occurred, leading to the improvement of Young’s modulus and tensile strength [34]; (3) graphene tube has smaller elongation at break compared with PDMS. Once the fracture of GNs happened, the graphene tube propagated either along the interface, or into the PDMS matrix to cause final failure, resulting in the decrease of elongation at break of PDMS [37].

We have also conducted 100 stretching and release cycles to 100% tensile strain on the composite, and the composite shows a moderate decrease (<13%) in the stress required to deform it to this strain level at the 100th cycle (Figure 6b), indicating that the composite is intact for up to 100 stretching and release cycles. This result can be further confirmed by the measured conductivity of the composite (5900 S/m), which shows slight degradation after 100 stretching and release cycles.

With the excellent electrical conductivity and mechanical properties, the PDMS/GN composite has good potential application in EMI Shielding. The EMI SE is an evaluation criterion of material’s ability to attenuate the power of electromagnetic waves during the electromagnetic waves transmitted through. The EMI SE of PDMS/GN composites and pristine PDMS in X-band (8–12 GHz) can be extracted from the measured the *S*-parameter of the composite according the following equation [32,33,34,35,36,37,38,39]:(1)$$\mathrm{EMI}\;\mathrm{SE}\;(\mathrm{dB})=10\log(P_{i}/P_{t})=-10\log(1/{\left|S_{21}\right|}^{2})$$
where *P_i_* and *P_t_* are the incident and transmitted power, respectively. As shown in Figure 7a, the pristine PDMS shows a calculated EMI SE of 1 dB, indicating that all the energy of an electromagnetic wave can almost be transported through the PDMS. However, the prepared PDMS/GN composite with the thickness of 0.25 mm has an EMI SE above 35 dB in the whole X-band. This result indicates that only 0.01% electromagnetic wave power can be transmitted. This enhancement of the shielding efficiency is clearly attributed to the presence of GN in the composite. 

The thickness of the shielding materials plays an important role in the EMI shielding performance. To investigate the influence of the thickness on the EMI shielding performance of PDMS/GN composite, 2 and 3 pieces of PDMS/GN composites with the thickness of 0.25 mm (1.2 wt % GN loading) were stacked and tested. The result in Figure 7a shows that samples with thickness of 0.5 mm and 0.75 mm exhibited an average EMI SE of ~60 dB and ~90 dB in the X-band, respectively. These values are the best results among the EMI SE of CPCs based on carbon filler, and comparable with CPCs with metals as the filler, as listed in Table 1. In addition, we have also investigated the stability of EMI SE of the composite after 100 stretching and release cycles, and only a slight degradation can be found (Figure 7b), indicating the composite is highly stable.

The EMI shielding of a material’s mechanisms depends mainly on two electromagnetic attenuation mechanisms, i.e., the wave reflection and absorption. The reflection relates to the reflected electromagnetic wave power on interface between the air and shield material, which results from the mismatched characteristic impedance; the absorption relates to the dissipative electromagnetic wave in the shield material caused by the interaction of the electromagnetic field and electric dipoles or magnetic dipoles [25,44,45,47]. Based on the above mechanisms, the incident electromagnetic wave power (*I*) reaching the shielding material shall be decomposed into three parts, i.e., the reflection (*R*) part, absorption (*A*) part and transmitted (*T*) part. *R*, *A*, *T* can be calculated by *S*-parameters [44,45,47]:(2)$$\begin{array}{l} R={\left|S_{11}\right|}^{2}={\left|S_{22}\right|}^{2}\\ T={\left|S_{12}\right|}^{2}={\left|S_{21}\right|}^{2}\\ A=1-{\left|S_{11}\right|}^{2}-{\left|S_{21}\right|}^{2}\\ I=R+A+T \end{array}$$

For the prepared PDMS/GN composites, the calculated *R*, *T* and *A* powers as a function of the thickness at 11.5 GHz are presented in Figure 8a. With increasing thickness, the reflection was enhanced while the absorption was degraded. For the 0.75 mm sample, 91.66% and 8.33% powers were reflected and absorbed, respectively, and only 0.0000004% was transmitted, indicating the superior EMI SE of the material. Based on this result, it can be concluded that most power was blocked by reflection, which can be attributed to the superior conductivity of the composites, leading to the significant mismatch between the materials and the air. However, this result cannot deduce the conclusion that the dominant shielding mechanism of the composite is reflection. In fact, the lower absorption is due to the lower power that enters the sample as a result of the mismatch. The contribution of absorption to the overall shielding should be based on the material’s ability in attenuating the power that has not been attenuated by reflection [44,45,47]. According to the previous reports [48,49], for the total power entering the samples, the total loss (*SE_T_*) and contribution of absorption loss (*SE_A_*) and reflection loss (*SE_R_*) can be calculated as follows:(3)$$\begin{array}{l} \begin{array}{l} SE_{R}=10log_{10}I/\left(I-R\right)\\ SE_{A}=-10log_{10}\left[T/\left(I-R\right)\right] \end{array}\\ SE_{T}=SE_{A}+SE_{R} \end{array}$$

Figure 8b shows that *SE_A_* and *SE_R_* increase with the increasing in electrical conductivity. It is worth noting that the contribution of absorption to the EMI shielding effectiveness is much larger than that of reflection. For the 0.75 mm composite, *SE_T_*, *SE_A_*, and *SE_R_* are ~85, 73 and 12 dB, respectively. After comparing the results in Figure 8a,b, it can be concluded that nanocomposite materials have high intrinsic absorption capabilities. However, since reflection takes place before absorption, most of the incident wave is reflected because of the mismatch as discussed above.

## 4. Conclusions

A flexible PDMS/GN composite was developed with an outstanding EMI shielding effectiveness and tensile strength. The 3D graphene network was prepared by a CVD method using a 3D nickel network as the template and subsequent etching process. The prepared 3D graphene network has a high electrical conductivity of 6300 S/m. The PDMS/GN composite produced with a vacuum infiltration method inherits the high electrical conductivity at a very low loading level of graphene (1.2 wt %). With this superior electrical conductivity, outstanding EMI shielding effectiveness of around 40, 60 and 90 dB can be achieved in the X-band when thicknesses of the composite are 0.25, 0.5 and 0.75 mm, respectively. In addition to the superior EMI SE, the 3D graphene tube network can also act as a mechanical enhancer for PDMS. 

## Figures and Tables

**Figure 1 nanomaterials-10-00768-f001:**
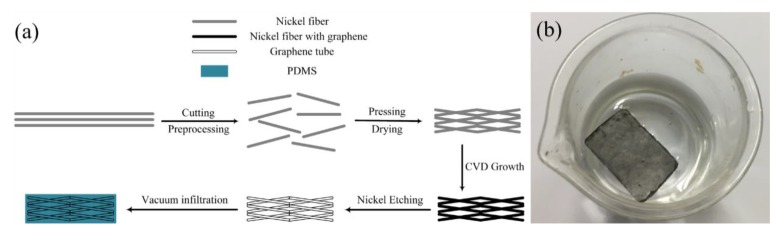
(**a**) Preparation process of porous polydimethylsiloxane/graphene network (PDMS/GN) nanocomposite; (**b**) prepared PDMS/GN floating on water.

**Figure 2 nanomaterials-10-00768-f002:**
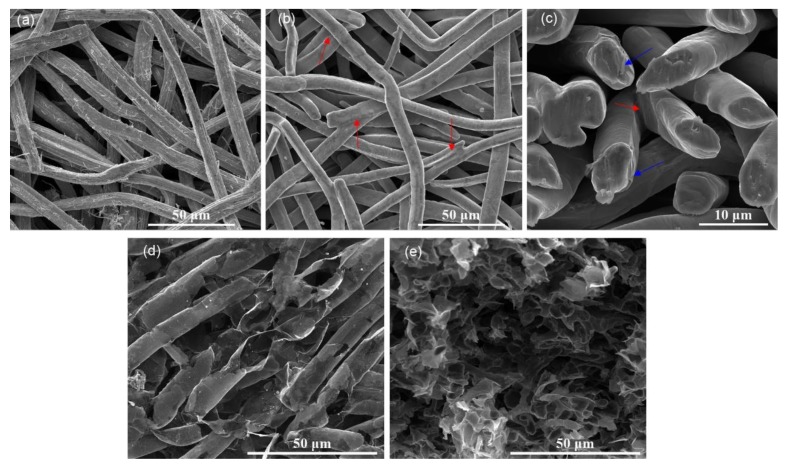
Top view scanning electron microscope (SEM) images of nickel network before (**a**) and after chemical vapor deposition (CVD) process (**b**); cross-section view of nickel network after CVD process (**c**); top view (**d**) and cross-section view (**e**) of GN after nickel was etched away. Red arrows in (**b**,**c**) indicate the nodes where nickel fibers were sintered and aggregated, blue arrows in (**c**) indicate the positions where graphene was peeled off.

**Figure 3 nanomaterials-10-00768-f003:**
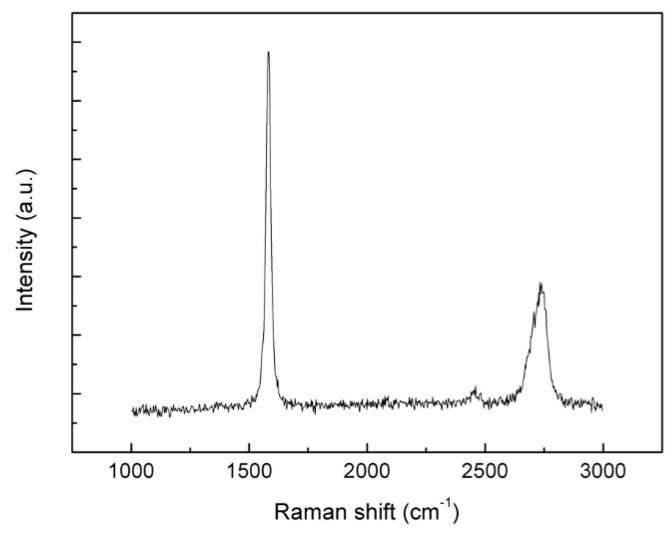
Raman spectra of the prepared GN.

**Figure 4 nanomaterials-10-00768-f004:**
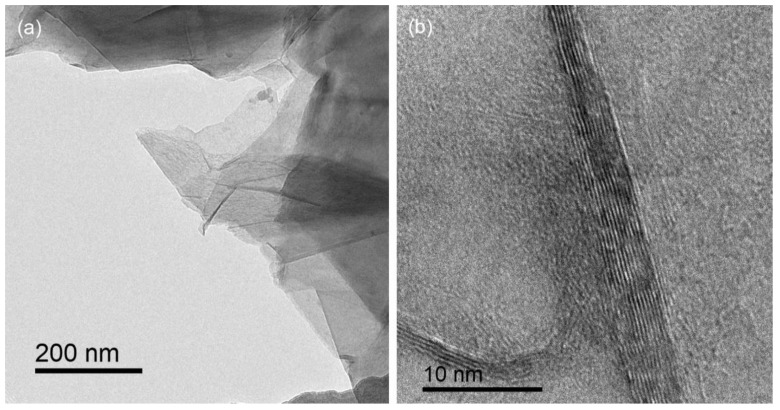
Transmission electron microscope (TEM) images of GN, indicating the (**a**) microstructure and (**b**) thickness.

**Figure 5 nanomaterials-10-00768-f005:**
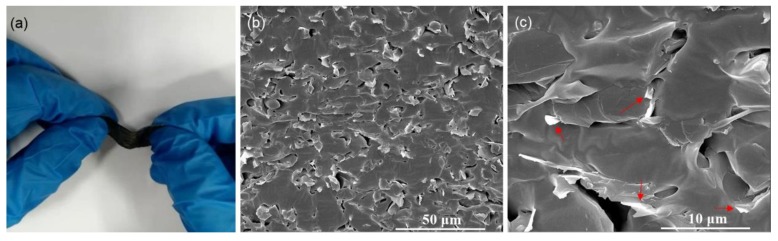
(**a**) A bending PDMS/GN composite; (**b**) and (**c**) SEM images of the PDMS/GN composite, the red arrows in (**c**) indicates the graphene.

**Figure 6 nanomaterials-10-00768-f006:**
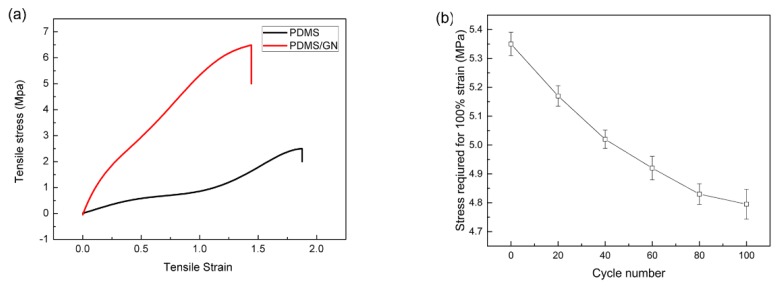
(**a**) Tensile strain-stress curves of PDMS and PDMS/GN; (**b**) stress required for 100% strain for 0–100 stretching and release cycles.

**Figure 7 nanomaterials-10-00768-f007:**
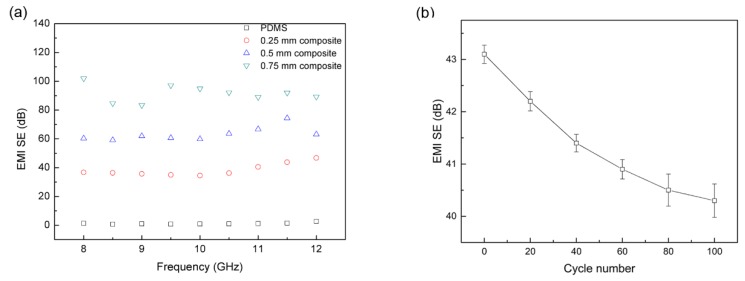
(**a**) Electromagnetic interference (EMI) shielding effectiveness (SE) of the PDMS/GN composite with different thickness in X-band; (**b**) EMI SE of the 0.25 mm composite at 11.5 GHz after stretching and release cycles for 0–100 cycles.

**Figure 8 nanomaterials-10-00768-f008:**
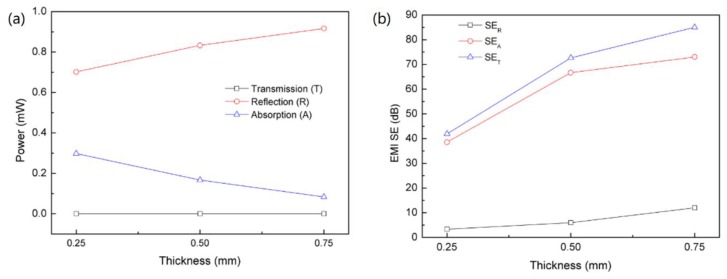
(**a**) Power balance of composites as function of thickness; (**b**) contribution of reflection and absorption to the EMI SE of composites as a function of thickness.

**Table 1 nanomaterials-10-00768-t001:** Comparison of electrical and EMI SE properties of various CPCs. (PE: polyethylene, PU: polyurethane, SWCNT: single wall carbon nano tube, Gr: graphene).

Polymer Matrix	Filler	Filler Loading (%)	Conductivity (S/m)	Thickness (mm)	Frequency	EMI SE (dB)	Ref.
Epoxy	r-GO	15 wt	~4	2	X-band	21	[38]
Epoxy	SWCNT	15 wt	0.2	2	X-band	25	[39]
PE	MWCNT	10 wt	-	1	X-band	50	[40]
PS	r-GO	7.0 wt	43.5	2.5	X-band	45.1	[41]
PS	S doped r-GO	7.5 vol	33	2	12–18 GHz	24.5	[42]
PS	MWCNT	20 wt	0.0072	2	X-band	63.3	[43]
PU	SWCNT	20 wt	0.00022	1	X-band	17	[44]
WPU	r-GO	7.7 wt	5.1	2	X-band	32	[45]
PU	Gr	20 wt	2500	1	X-band	80	[31]
PDMS	GN	0.8 wt	200	1	X-band	19.98	[33]
PDMS	MWCNT	5.7 vol	301	2	X-band	80	[46]
PDMS	GN	1.2 wt	6100	0.75	X band	90	Present work
silicone	Ag + Al	/	12,500	/	10 GHz	85	Commercial products (Holland Shielding Systems B.V.)

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
