# Peer review of "Highly Conductive PDMS Composite Mechanically Enhanced with 3D-Graphene Network for High-Performance EMI Shielding Application"

_nanomaterials, 2020, doi:10.3390/nano10040768_

Round 1
Reviewer 1 Report
In this study by Ao et al. the authors have demonstrated shielding properties of a polymer filled with graphene. The results are interetsing and fit a modern topic, but several issues must be addressed first to warrant publication of this work. Please see my suggestions below:
1) The level of English should be significantly improved as presently it is very hard to understand the meaning you are trying to convey.
2) State of the art should be updated with results on EMI shielding using other types of nanocarbon such as carbon nanotubes. There is a lot of results in this topic e.g. 10.1016/j.carbon.2017.09.078
3) Some figures e.g. Fig. 1a are distorted. Please correct it.
4) The expression "graphene tube" is quite misleading. That suggests a hollow structure and it is not tubular at all especially after Ni removal. Please consider another way of referring to your material.
5) Professional scale bar markers should be included in SEM micrographs.
6) Your data does not contain any error bars. How many samples were tested for each parameter set?
Reviewer 2 Report
The manuscript is interesting and the research is novel and highly relevant to researchers working on this topic. I have the following comments:
Minor comments:
1. The authors use the acronym EMI SE without defining SE, from reading the article, I am guessing SE stands for shielding effectiveness. Please define the acronym when it first appears in the abstract.
2. On line 193-194, "With the excellent electrical conductivity and mechanical properties, the PDMS/GN 193 composite is highly potential in EMI Shielding application", there seems to be a typo, is highly potential should probably be has high potential...
Major comments:
1. How was the mechanical properties of the composite being measured? The authors indicated that 100 stretching and releasing cycle was performed, how was the cycle being performed exactly? Please clarify.
2. Similarly how was the conductivity of the composite being measured?
3. How does the composite material behaved under power loading? That is the author mentioned that the incident power used was 1 mW, how high of a power can the material handle before breaking down?
Reviewer 3 Report
The work can be accepted in its present form. Nice work!
Author Response
Dear Reviewer, Thank you for your attention and careful reviews on our manuscript nanomaterials-749341. According to reviewers' suggestions, we have made a careful revision on the previous version of the manuscript, and these changes will not influence the content and framework of this paper. Revised portion are marked in red (rewritten) or yellow (added) in the revised manuscript. At last, thank you for your reviewing again.Round 2
Reviewer 1 Report
Revisions conducted satisfactorily. Please accept the work for publication.